# Deep Learning of the Biswas–Chatterjee–Sen Model

**DOI:** 10.3390/e27111173

**Published:** 2025-11-20

**Authors:** José F. S. Neto, David S. M. Alencar, Lenilson T. Brito, Gladstone A. Alves, Francisco Welington S. Lima, Antônio M. Filho, Ronan S. Ferreira, Tayroni F. A. Alves

**Affiliations:** 1Departamento de Matemática e Física, Universidade Estadual do Maranhão, Caxias 65604-380, MA, Brazil; ferreiraesol@gmail.com (J.F.S.N.); d.s.m.alencar@gmail.com (D.S.M.A.); 2Departamento de Física, Universidade Estadual do Piauí, Teresina 64002-150, PI, Brazil; lenilsontorres@ccn.uespi.br (L.T.B.); alves.gladstone@gmail.com (G.A.A.); amfilho@gmail.com (A.M.F.); 3Departamento de Física, Universidade Federal do Piauí, Teresina 57072-970, PI, Brazil; fwslima@gmail.com; 4Departamento de Ciências Exatas e Aplicadas, Universidade Federal de Ouro Preto, João Monlevade 35931-008, MG, Brazil; ronan.ferreira@ufop.edu.br

**Keywords:** deep learning, supervised learning, unsupervised learning, non-equilibrium phase transition, consensus formation, kinetic continuous opinion dynamics

## Abstract

We investigate the critical properties of kinetic continuous opinion dynamics using deep learning techniques. The system consists of *N* continuous spin variables in the interval [−1,1]. Dense neural networks are trained on spin configuration data generated via kinetic Monte Carlo simulations, accurately identifying the critical point on both square and triangular lattices. Classical unsupervised learning with principal component analysis reproduces the magnetization and allows estimation of critical exponents. Additionally, variational autoencoders are implemented to study the phase transition through the loss function, which behaves as an order parameter. A correlation function between real and reconstructed data is defined and found to be universal at the critical point.

## 1. Introduction

Machine learning (ML) comprises a set of techniques for analyzing large volumes of data and is now a key tool in diverse fields, including statistics, condensed matter, high energy physics, astrophysics, cosmology, and quantum computing. ML encompasses three main approaches: supervised learning (using labeled data to learn mappings and make predictions), unsupervised learning (discovering patterns without labels, such as clustering and dimensionality reduction), and reinforcement learning (where an agent learns optimal policies through rewards and penalties).

In this work, we apply supervised and unsupervised deep learning methods to investigate disorder-induced phase transitions in the Biswas–Chatterjee–Sen (BChS) model. In this model, opinions are continuous, $s_{i}\in \left[-1,1\right]$, and pairwise interactions can be cooperative (+) or antagonistic (−). The fraction *q* of antagonistic interactions controls the disorder. On fully connected networks, the model exhibits a continuous phase transition with Ising mean-field exponents [1].

Various geometries have been explored in the literature, motivating our analysis. On regular lattices, as for example square and cubic, the continuous version exhibits a second-order transition and belongs to the Ising universality class in the corresponding dimensions [2]. On Solomon networks (two coupled networks), both discrete and continuous versions show a continuous transition. In Solomon networks, the exponents depend on dimensionality and may differ from those of the Ising model [3,4,5].

On Barabási–Albert networks, the discrete version exhibits a second-order transition and universality class differences compared to other topologies [6]. On random and complex graphs, extensions with memory and bias confirm the occurrence of transitions and discuss changes in universality class [7]. Modular structures with two groups reveal, in the mean-field regime, a stable antisymmetric ordered state in addition to symmetric ordered and disordered states [8].

The aim of this work is to employ deep learning methods to investigate the continuous phase transition in the BChS model, demonstrating the applicability of these techniques to various network geometries. We generate data using kinetic Monte Carlo dynamics and analyze the resulting configurations with dense neural network classifiers, principal component analysis (*PCA*), and variational autoencoders (*VAE*) to accurately identify the critical point and characterize the critical behavior, even in the presence of disorder.

In the following sections, we apply machine learning techniques to study the continuous phase transition of the BChS model on both square and triangular lattices. We begin by presenting our results using supervised learning with dense neural networks, followed by unsupervised learning with *PCA* and dense neural networks.

## 2. Data Generation

We generate spin configuration data for the BChS model using the following kinetic rules [1,2]:Assign to each of the *N* nodes in the network an opinion variable $s_{i}$ in the continuous interval [−1,1]. The network state is(1)$$s=\left(s_{1},s_{2},\ldots ,s_{N}\right).$$The initial configuration is generated by randomly sampling each $s_{i}$ from a uniform distribution in [−1,1].At each step (discrete time), randomly select a node *i* for update.Randomly select a neighbor *j* of node *i*. The affinity parameter $\mu_{i,j}$ for the link between *i* and *j* is chosen randomly: $\mu_{i,j}$ is an annealed random variable in [−1,1], negative with probability *q* (antagonistic interaction), positive with probability 1−q (cooperative interaction).Update both nodes *i* and *j* according to(2)$$\left\{\begin{array}{c} s_{i}\left(t+1\right)=s_{i}\left(t\right)+\mu_{i,j}s_{j}\left(t\right),\\ s_{j}\left(t+1\right)=s_{j}\left(t\right)+\mu_{i,j}s_{i}\left(t\right), \end{array}\right.$$
where $s_{i}\left(t\right)$ and $s_{j}\left(t\right)$ are the states before the update, and $s_{i}\left(t+1\right)$, $s_{j}\left(t+1\right)$ are the updated states. One Monte Carlo step (MCS) consists of *N* such updates.If any updated state $s_{i,j}\left(t+1\right)$ exceeds 1, set $s_{i,j}\left(t+1\right)=1$; if $s_{i,j}\left(t+1\right)<-1$, set $s_{i,j}\left(t+1\right)=-1$. This enforces the bounds and introduces nonlinearity.

The BChS model exhibits a continuous phase transition at a critical noise $q_{c}$ between a ferromagnetic phase ($q<q_{c}$) with nonzero average opinion and a paramagnetic phase ($q>q_{c}$) with zero average opinion.

To collect stationary configurations, we discard the first $N_{\mathrm{term}}=10^{4}$ Monte Carlo steps. After the dynamics become stationary, we sample $N_{t}$ configurations $s_{l}$ ($l=0,1,\ldots ,N_{t}$), discarding additional steps between samples to reduce correlations [9]. These stationary configurations are then used for deep learning analysis of the BChS model on square and triangular lattices.

## 3. Supervised Learning

Neural networks have been widely applied to study second-order phase transitions in models such as the Ising model [10,11,12], directed percolation [13], the pair contact process with diffusion [14], and quantum phase transitions [15]. In this work, we extend these methods to the BChS model on square and triangular lattices. Dense neural networks are trained to classify configurations as ferromagnetic ($q<q_{c}$) or paramagnetic ($q>q_{c}$). Training is performed on square lattice data, and inference is carried out on stationary configurations from both square and triangular lattices, allowing us to assess the network’s ability to identify the critical point in a nonequilibrium system with continuous states.

To account for the $Z_{2}$ symmetry, each configuration generated during simulation is paired with its inverted counterpart. The resulting training dataset contains $N_{D}=4\times 10^{6}$ configurations, comprising $10^{4}$ stationary samples for each of 200 noise values in the range $0.5q_{c}^{s}$ to $1.5q_{c}^{s}$ on the square lattice. The critical noise for the BChS model on the square lattice is $q_{c}^{s}\approx 0.2266$ [2]. Of the total dataset, 20% is reserved for validation.

During Monte Carlo simulations, the first $10^{4}$ steps are discarded to ensure that the dynamics become stationary, and an additional $10^{3}$ steps are omitted between stored configurations to reduce correlations. One Monte Carlo step corresponds to updating all $N=L^{2}$ spins. Lattice sizes used are L=16, 20, 24, 32, and 40. Configurations sampled at $q<q_{c}^{s}$ are labeled as ferromagnetic, while those at $q>q_{c}^{s}$ are labeled as paramagnetic.

The neural network architecture is as follows:Input layer of size $L^{2}$, with each input representing a continuous spin value $s_{i}\in \left[-1,1\right]$;First hidden layer with 128 neurons, ReLU activation, $l_{2}$ regularization, batch normalization, and dropout rate 0.2;Second hidden layer with 64 neurons, ReLU activation, $l_{2}$ regularization, batch normalization, and dropout rate 0.2;Output layer with two neurons ($\rho_{1}$, $\rho_{2}$) and softmax activation.

The output $\rho_{1}$ represents the score for a pure ferromagnetic state (q=0), while $\rho_{2}$ is the complementary score for a paramagnetic state (q→∞). Configuration labels are set as $y_{i}=1$ for the ferromagnetic phase and $y_{i}=0$ for the paramagnetic phase. The point of maximum confusion, corresponding to the transition threshold, occurs when $\rho_{1}=\rho_{2}=0.5$. We chose a dense neural network architecture for this task, as dense neural networks are well suited for classification problems on arbitrary geometries since they do not rely on spatial structure. The neural network is implemented and trained using the Keras 3.11.3 and Tensorflow 2.20.0 libraries in Python 3.13.5.

The neural network was trained for at least $10^{3}$ epochs with a batch size of 128, using the ADAM 3.11.3 optimizer with a learning rate $\eta =10^{-4}$, and the chosen loss function is the sparse categorical cross-entropy(3)$$l_{\mathrm{SCE}}=-\frac{1}{N_{D}}\sum_{i=1}^{N_{D}}y_{i}\ln y_{i}^{\prime }\left(\theta \right),$$
where $N_{D}$ is the size of the dataset, $y_{i}$ is the true label of the configuration, $y_{i}^{\prime }\left(\theta \right)$ is the predicted label by the neural network, and θ represents the neural network parameters (weights and biases), which are optimized during training. The categorical cross-entropy loss function measures the dissimilarity between the true and predicted labels, encouraging the neural network to make accurate classifications.

Figure 1 shows the classification results for the BChS model on the square lattice. In panel (a), the crossing point $\rho_{1}=\rho_{2}=0.5$ closely matches the critical noise $q_{c}^{s}$, indicated by the dashed vertical line. Panel (b) demonstrates that the outputs collapse according to the finite-size scaling relation(4)$$\rho_{1,2}\propto f_{\rho_{1,2}}\left(N^{1/\nu }\left(q-q_{c}^{\prime }\right)\right),$$
where ν=1 is the correlation length exponent for the Ising universality class in two dimensions, and $q_{c}^{\prime }$ denotes the crossing abscissas. The scaling functions $f_{\rho_{1}}$ and $f_{\rho_{2}}$ are universal up to a rescaling of the argument. These results confirm that the neural network accurately identifies the critical point $q_{c}^{s}$ of the BChS model on the square lattice.

Next, we generated an inference dataset for the triangular lattice using the same parameters as for the square lattice. The neural network trained on square lattice data was then used to infer on triangular lattice configurations. The results are shown in Figure 2. In panel (a), the crossing point $\rho_{1}=\rho_{2}=0.5$ is close to the critical noise $q_{c}^{t}$, indicated by the dashed vertical line. Panel (b) shows that the outputs collapse according to Equation (Equation 4) with ν=1.

An extrapolation of the crossing points $q_{c}^{\prime }$ seen in Figure 2 as a function of 1/L is shown in Figure 3. The extrapolation is performed according to the linear regression(5)$$q_{c}^{\prime }=q_{c}-\frac{a}{L},$$
where *a* is a constant. The extrapolation yields an estimate for the critical noise, $q_{c}\approx 0.2397\pm 0.0002$, which provides an estimate of the critical noise of the BChS model on the triangular lattice.

Results for the BChS model on the triangular lattice are not available in the literature. To compare the classification neural network results with standard methods, we simulated the model on triangular lattices using the kinetic Monte Carlo method. The fundamental observable is the average opinion balance (magnetization) per spin:(6)$$m=\left|\frac{1}{N}\sum_{i=1}^{N}s_{i}\right|.$$
The order parameter is the time average of *m* in the stationary regime, and its fluctuation defines the susceptibility. The order parameter *M*, susceptibility χ, and Binder cumulant *U* are defined as [16](7)$$\begin{array}{ccc} M(q) & = & \langle m\rangle ,\\ \chi (q) & = & N\left(\langle m^{2}\rangle -{\langle m\rangle }^{2}\right),\\ U(q) & = & 1-\frac{\langle m^{4}\rangle }{3{\langle m^{2}\rangle }^{2}}, \end{array}$$
where 〈⋯〉 denotes the time average over the Markov chain. All observables depend on the noise parameter *q*, so independent simulations are performed for each value of *q*.

We performed simulations on triangular lattices of sizes L=50, 60, 70, 80, 90, and 100. For each noise value, we discarded the first $2\times 10^{5}$ Monte Carlo steps to ensure stationarity, then collected $10^{7}$ samples, omitting 10 steps between samples to reduce correlations. The results are shown in Figure 4. We estimated the critical noise $q_{c}^{t}\approx 0.240$ using Binder’s cumulant method [17], which is close to the extrapolation estimate shown in Figure 3. The critical behavior matches that of the square lattice, as expected, except for the non-universal value of the critical noise. The agreement between the critical noise estimated by the neural network and that obtained via Monte Carlo simulations confirms the effectiveness of supervised learning in identifying phase transitions in nonequilibrium systems with continuous states.

## 4. Unsupervised Learning

Unsupervised learning methods have also been applied to study phase transitions in the Ising model [18,19,20]. Here, we extend both supervised and unsupervised learning approaches to the BChS model on square and triangular lattices. We first employ *PCA*, a classical unsupervised technique, followed by *VAEs*, a deep learning method.

PCA identifies the directions (principal components) along which the variance in the data is maximized. The first principal component captures the largest variance, the second captures the next largest, and so on. These components correspond to the eigenvectors of the covariance matrix, with their associated eigenvalues indicating the amount of variance explained. *PCA* is commonly used for dimensionality reduction, visualization, and feature extraction.

Figure 5 shows the results of *PCA* applied to BChS model training data. For low noise values, the principal components form two clusters centered at (0,−L) and (0,L). For higher noise values, a single cluster appears at (0,0). The cluster plot provides a clear visualization of the phase transition in the principal component space.

We further analyzed the principal component data using finite-size scaling techniques. Specifically, we considered the ratio of the two largest eigenvalues $\lambda_{2}/\lambda_{1}$ of the covariance matrix, as well as the averages of the absolute values of the first and second principal components, denoted $P_{1}$ and $P_{2}$, respectively. The finite-size scaling relations for these observables are(8)$$\begin{array}{ccc} \lambda_{2}/\lambda_{1} & \propto & f_{\lambda }\left(N^{1/\nu }\left(q-q_{c}\right)\right),\\ P_{1}/L & \propto & N^{-\beta /\nu }f_{P_{1}}\left(N^{1/\nu }\left(q-q_{c}\right)\right),\\ LP_{2} & \propto & N^{\gamma /\nu }f_{P_{2}}\left(N^{1/\nu }\left(q-q_{c}\right)\right), \end{array}$$
where $f_{\lambda }$, $f_{P_{1}}$, and $f_{P_{2}}$ are universal scaling functions.

Figure 6 summarizes the results for these *PCA* observables. Panel (a) shows that the ratio $\lambda_{2}/\lambda_{1}$ is universal at the critical noise $q_{c}^{s}$ for the square lattice. Panel (b) demonstrates the scaling collapse, allowing estimation of the critical exponent ν=1. Panel (c) presents $P_{1}/L$ as a function of noise, which coincides with the average magnetization per spin and scales as $L^{\beta /\nu }$ with β/ν=1/8, as confirmed by the collapse in panel (d). Panel (e) displays $LP_{2}$, whose maximum increases with $L^{\gamma /\nu }$ at the critical point, where γ/ν=7/4, as shown in panel (f).

We also investigated the continuous phase transition using *VAEs*, which are generative models combining autoencoder architectures with variational inference. A *VAE* consists of an encoder that maps input data to a latent space and a decoder that reconstructs the input from the latent representation. The encoder learns a probabilistic mapping, enabling the generation of new samples by sampling from the latent space.

The encoder architecture is as follows:Input layer of size $L^{2}$, with each input corresponding to a continuous spin variable $s_{i}\in \left[-1,1\right]$;First hidden layer with 625 neurons, ReLU activation, $l_{1}$ regularization, batch normalization, and dropout rate 0.2;Second hidden layer with 256 neurons, ReLU activation, $l_{1}$ regularization, batch normalization, and dropout rate 0.2;Third hidden layer with 64 neurons, ReLU activation, $l_{1}$ regularization, batch normalization, and dropout rate 0.2;Output layer with two neurons (linear activation): one outputs the mean μ and the other outputs the logarithm of the variance σ of the latent variable *Z*.

The decoder mirrors the encoder structure and receives the latent encoding *Z* as input. Additionally, it includes an extra input neuron for the normalized noise of the configuration, making the neural network a conditional *VAE*.

The *VAE* was trained for at least $10^{3}$ epochs with a batch size of 128, using the RMSprop optimizer with a learning rate $\eta =10^{-3}$. The loss function is the sum of the mean squared error and the Kullback–Leibler loss(9)$$l_{\mathrm{VAE}}=l_{\mathrm{MSE}}+l_{\mathrm{KL}},$$
where $l_{\mathrm{MSE}}$ is the mean squared error between the input and reconstructed configurations,(10)$$l_{\mathrm{MSE}}=\frac{1}{N_{D}}\sum_{i=1}^{N_{D}}{\left[y_{i}-y_{i}^{\prime }\left(\theta \right)\right]}^{2},$$
with $N_{D}$ the dataset size, $y_{i}$ the input configuration, $y_{i}^{\prime }\left(\theta \right)$ the reconstructed configuration, and θ the network parameters. The Kullback–Leibler loss [21] is(11)$$l_{\mathrm{KL}}=-\frac{1}{2}\sum_{i=1}^{d}\left[1+\log\sigma_{i}^{2}-\mu_{i}^{2}-\sigma_{i}^{2}\right],$$
where *d* is the dimension of the latent space, and $\mu_{i}$, $\sigma_{i}$ are the mean and standard deviation of the latent variable. The Kullback–Leibler loss regularizes the latent space by encouraging the learned distribution to approximate a standard normal distribution, preventing overfitting and enabling sampling of new artificial configurations.

The latent space consists of a single statistical variable *Z*, sampled from a normal distribution with mean μ and variance σ provided by the encoder. This minimal latent space encourages the encoder to capture only the most relevant features of the data and prevents trivial reproduction of the input configurations. The *VAE* was implemented and trained using the Keras and Tensorflow libraries in Python.

Figure 7 shows the latent encoding *Z* of the input data as a function of magnetization and normalized noise. In panel (a), a clear separation between positive and negative magnetizations is observed according to the sign of *Z*, reflecting the $Z_{2}$ symmetry. Panel (b) demonstrates that the relationship learned by the neural network between magnetization *m* and latent encoding *Z* is approximately linear. At low noise values, two clusters centered at (−2,−1) and (2,1) appear, while at higher noise values, a single cluster at (0,0) emerges, indicating the phase transition. Panel (c) further confirms that the phase transition is evident from the latent encoding.

We define a correlation function between the real data configurations $s_{\mathrm{real}}$ and those reconstructed by the *VAE*, $s_{\mathrm{recon}}$, as(12)$$C\left(s_{\mathrm{real}}|s_{\mathrm{recon}}\right)\equiv \frac{1}{L^{2}}\frac{\langle \left|s_{\mathrm{real}}\cdot s_{\mathrm{recon}}\right|\rangle }{m_{\mathrm{real}}m_{\mathrm{recon}}},$$
where 〈⋯〉 denotes an average over the dataset, $m_{\mathrm{real}}$ is the average magnetization of the real data, and $m_{\mathrm{recon}}$ is the average magnetization of the reconstructed data. The correlation function is universal at the critical point, enabling estimation of the transition threshold in the same way as the Binder cumulant in Monte Carlo simulations. Therefore one can expect the following scaling dependence(13)$$C\left(s_{\mathrm{real}}|s_{\mathrm{recon}}\right)\propto f_{C}\left(N^{1/\nu }\left(q-q_{c}\right)\right),$$
which allows estimation of the correlation length exponent ν.

We also calculate the binary cross-entropy loss function $l_{\mathrm{BCE}}$,(14)$$l_{\mathrm{BCE}}=-\sum_{i=1}^{N_{D}}y_{i}\ln y_{i}^{\prime }\left(\theta \right)-\left(1-y_{i}\right)\ln\left[1-y_{i}^{\prime }\left(\theta \right)\right],$$
by renormalizing the input configurations to the interval [0,1]. The loss functions $l_{\mathrm{MSE}}$ and $l_{\mathrm{BCE}}$ between the input and reconstructed output configurations serve as indicators of the phase transition. In the paramagnetic regime (T→∞), the input and reconstructed outputs behave as two effectively random configurations, yielding limiting values $l_{\mathrm{MSE}}\to 3/2$ and $l_{\mathrm{BCE}}\to \ln2$ for random uniform data. Consequently, the quantities $1-3l_{\mathrm{MSE}}/2$ and $1-l_{\mathrm{BCE}}/\ln2$ act as order parameters and obey the following scaling relations:(15)$$\begin{array}{ccc} 1-3l_{\mathrm{MSE}}/2 & \propto & L^{2\beta /\nu }f_{\mathrm{MSE}}\left(N^{1/\nu }\left(q-q_{c}\right)\right),\\ 1-l_{\mathrm{BCE}}/\ln2 & \propto & L^{2\beta /\nu }f_{\mathrm{BCE}}\left(N^{1/\nu }\left(q-q_{c}\right)\right), \end{array}$$
where the loss functions scale with system size as 2β/ν=1/4, consistent with the universality class of the two-dimensional Ising model.

Figure 8 presents the *VAE* observables for the BChS model. Panel (a) shows the correlation function defined in Equation (Equation 12), which is universal at the critical noise $q_{c}^{s}$ for the square lattice. The scaling collapse in panel (b) confirms the finite-size scaling relation for the correlation function with the Ising critical exponent ν=1. Panels (c) and (e) display $3l_{\mathrm{MSE}}/2$ and $l_{\mathrm{BCE}}/\ln2$, respectively, both serving as order parameters that vanish at the transition. The scaling collapses in panels (d) and (f) confirm the expected scaling relations for these quantities with the Ising exponent 2β/ν=1/4.

## 5. Conclusions

In this work, we applied supervised and unsupervised deep learning techniques to study the continuous phase transition of the BChS model on square and triangular lattices. We generated spin configuration data using kinetic Monte Carlo simulations and trained dense neural networks to classify configurations into ferromagnetic and paramagnetic phases. The networks accurately identified the critical points, with outputs collapsing according to finite-size scaling relations.

Also, we employed *PCA* to analyze the data, revealing clustering behavior that visualizes the phase transition. The ratio of the two largest eigenvalues of the covariance matrix was found to be universal at the critical point, and we estimated critical exponents consistent with the Ising universality class. Furthermore, we implemented *VAEs* to study the phase transition through the loss function, which behaved as an order parameter. We defined a correlation function between the input and reconstructed configurations, finding it to be universal at the critical point. The scaling collapses of the correlation function and loss functions confirmed the critical exponents of the Ising universality class.

## Figures and Tables

**Figure 1 entropy-27-01173-f001:**
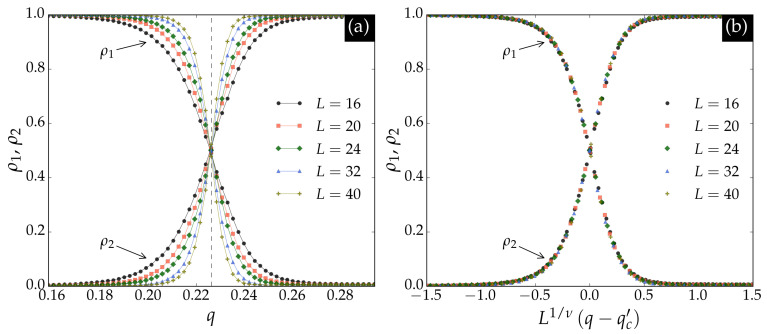
Neural network outputs for the BChS model on the square lattice. For each *L*, two curves are shown: $\rho_{1}$ and $\rho_{2}$. The score of ferromagnetic phase $\rho_{1}$ is close to 1 at low noise values and decreases at high noise values, while the score of paramagnetic phase $\rho_{2}$ behaves oppositely. The crossing of $\rho_{1}$ and $\rho_{2}$ marks the point of maximum confusion, corresponding to the transition threshold. In panel (**a**), the crossing point $\rho_{1}=\rho_{2}=0.5$ closely matches the critical noise $q_{c}^{s}$, indicated by the dashed vertical line. In panel (**b**), the outputs collapse according to Equation (Equation 4) with the critical exponent ν=1 for the square lattice; $q_{c}^{\prime }$ denotes the crossing abscissas.

**Figure 2 entropy-27-01173-f002:**
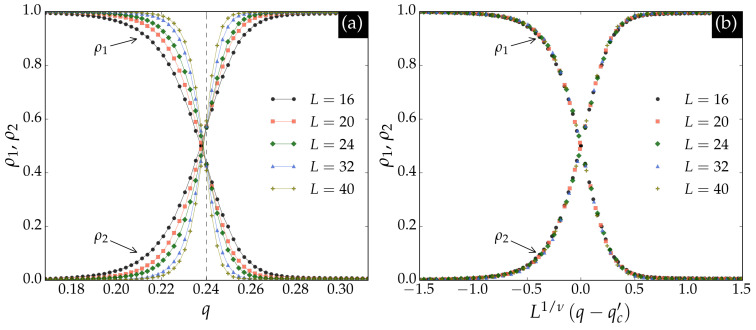
Neural network outputs $\rho_{1}$ and $\rho_{2}$ for the BChS model on the triangular lattice, trained with square lattice data. The curves have the same interpretation as in Figure 1. In panel (**a**), the crossing points $q_{c}^{\prime }$ ($\rho_{1}=\rho_{2}=0.5$) are used to estimate the critical noise via the process in Figure 3. The critical noise $q_{c}^{t}$ is indicated by the dashed vertical line. In panel (**b**), the outputs scale according to Equation (Equation 4) with critical exponent ν=1.

**Figure 3 entropy-27-01173-f003:**
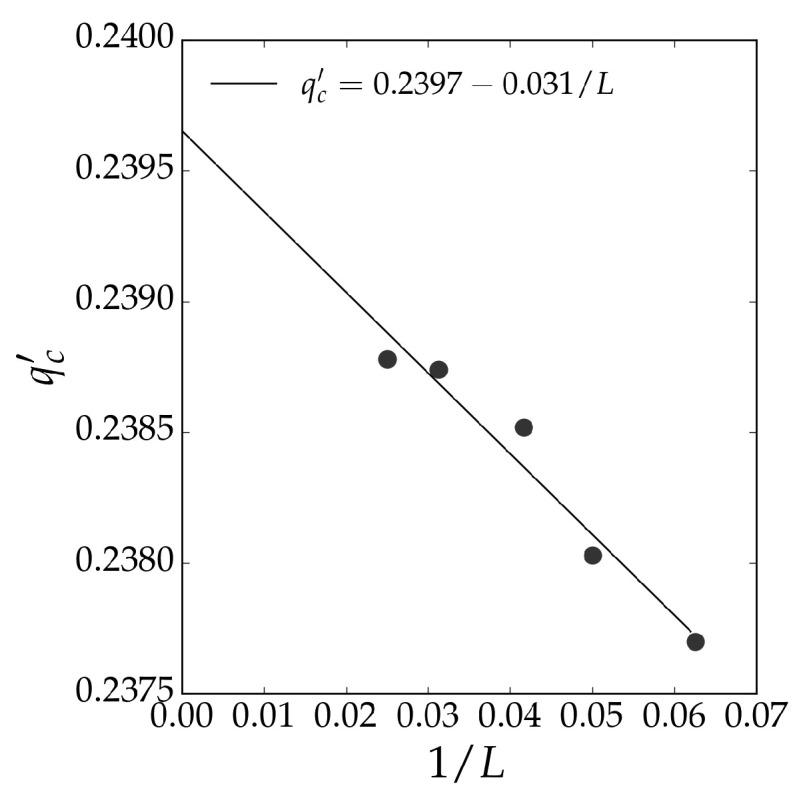
Linear regression of the crossing points $q_{c}^{\prime }$ of the neural network outputs $\rho_{1}$ and $\rho_{2}$ for BChS model configurations on the triangular lattice. Extrapolation according to Equation (Equation 5) yields an estimate for the critical noise, $q_{c}\approx 0.2397\pm 0.0002$. The black circles represent the critical noise for each network size.

**Figure 4 entropy-27-01173-f004:**
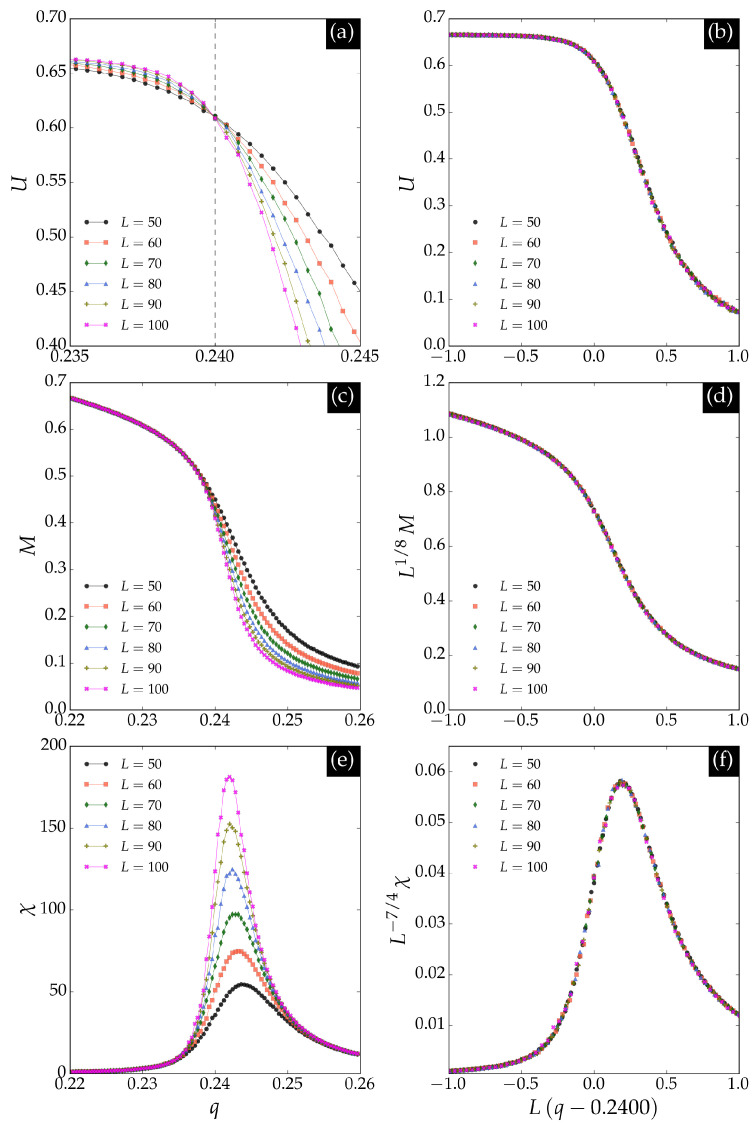
Observables for the BChS model on the triangular lattice obtained from standard Monte Carlo simulations. Panel (**a**) Binder cumulant *U* as a function of noise for different lattice sizes *L*. The curves intersect at the critical noise $q_{c}^{t}\approx 0.240$, indicated by the dashed vertical line. Panel (**b**) scaling transformation allows estimation of the critical exponent ν=1. Panel (**c**) order parameter *M* as a function of noise, which scales with $L^{\beta /\nu }$ where β/ν=1/8, as shown in panel (**d**). Panel (**e**) susceptibility χ, whose maximum increases with $L^{\gamma /\nu }$ at the critical point where γ/ν=7/4, as shown in panel (**f**).

**Figure 5 entropy-27-01173-f005:**
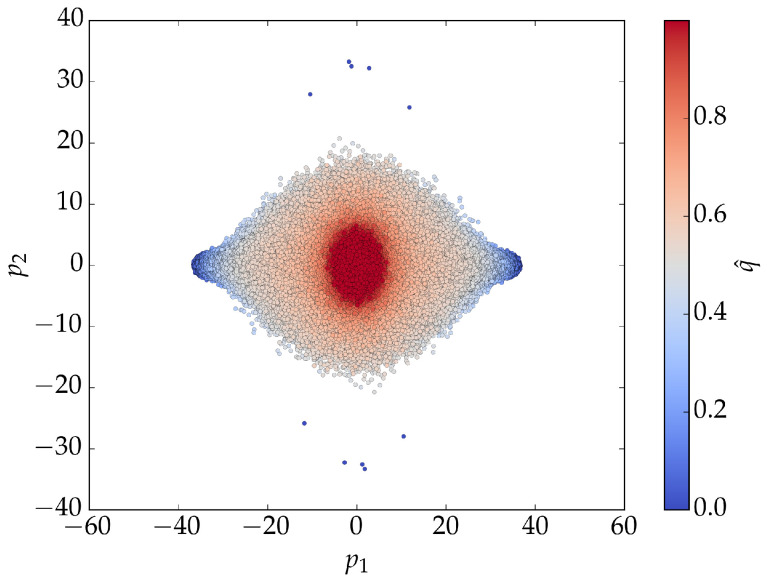
Projection of BChS model training data with L=40 onto the first two principal components as a function of the noise. *PCA* was performed separately for each noise value in the training set; $\hat{q}$ denotes normalized noise values from $0.5q_{c}^{s}$ to $1.5q_{c}^{s}$. For low noise, two clusters appear at (0,−L) and (0,L); for high noise, a single cluster emerges at (0,0). The clustering illustrates the phase transition.

**Figure 6 entropy-27-01173-f006:**
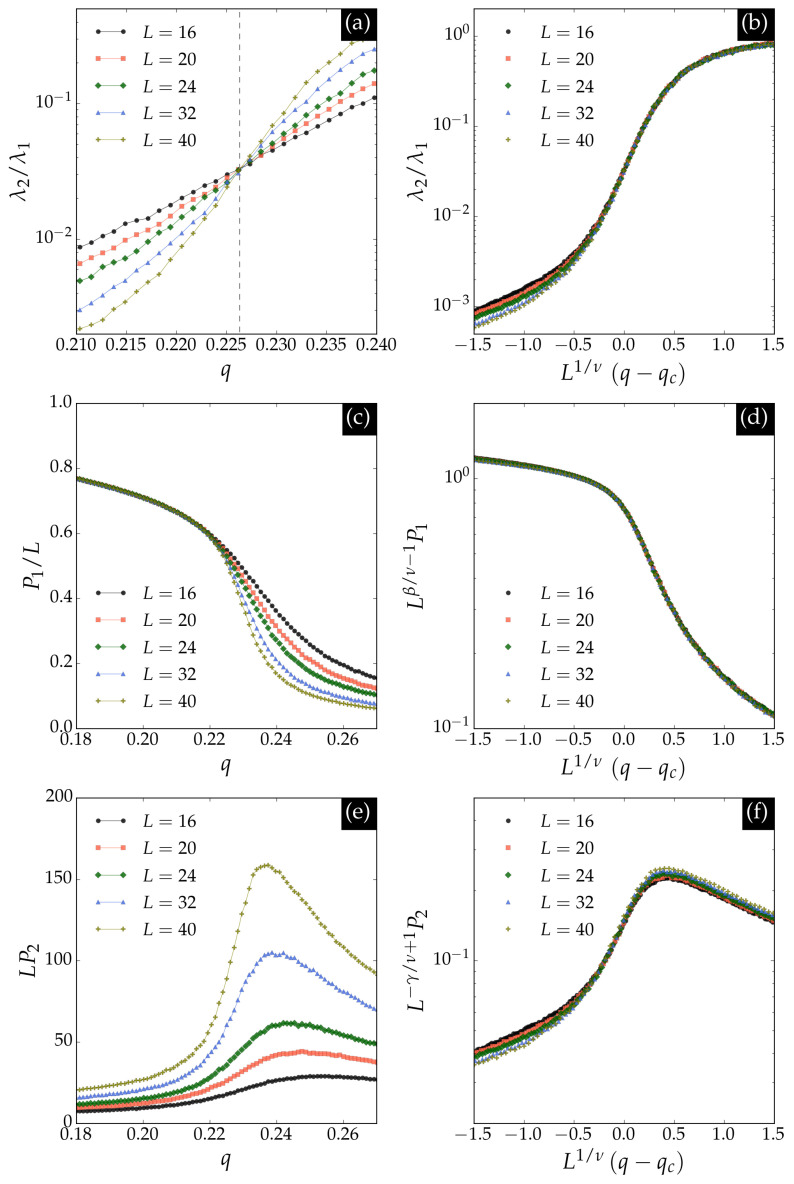
PCA observables. (**a**) Ratio of the two largest eigenvalues $\lambda_{2}/\lambda_{1}$, which is universal at the critical noise $q_{c}^{s}$. (**b**) Scaling collapse for $\lambda_{2}/\lambda_{1}$ with exponent ν=1. (**c**) $P_{1}/L$ as a function of noise, which coincides with the magnetization. (**d**) Scaling collapse for $P_{1}/L$ with β/ν=1/8. (**e**) $LP_{2}$ as a function of noise, whose maximum increases with $L^{\gamma /\nu }$ at the critical point. (**f**) Scaling collapse for $LP_{2}$ with γ/ν=7/4.

**Figure 7 entropy-27-01173-f007:**
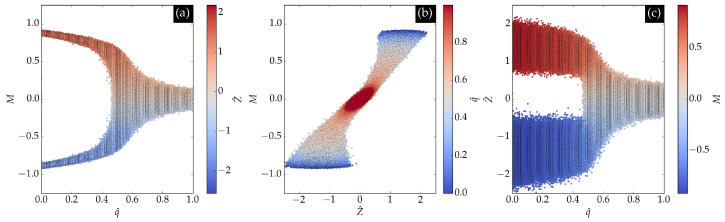
Dependence of the normalized latent encoding $\hat{Z}$ on magnetization and noise for BChS model data. Panel (**a**) magnetizations of input configurations as a function of normalized noise from $0.5q_{c}^{s}$ to $1.5q_{c}^{s}$, with the color gradient representing the latent encoding *Z*. Both magnetizations and encodings exhibit the $Z_{2}$ inversion symmetry. Panel (**b**) one can observe the nearly linear relationship between magnetization and latent encoding. Panel (**c**) the phase transition is also evident from the latent encoding data.

**Figure 8 entropy-27-01173-f008:**
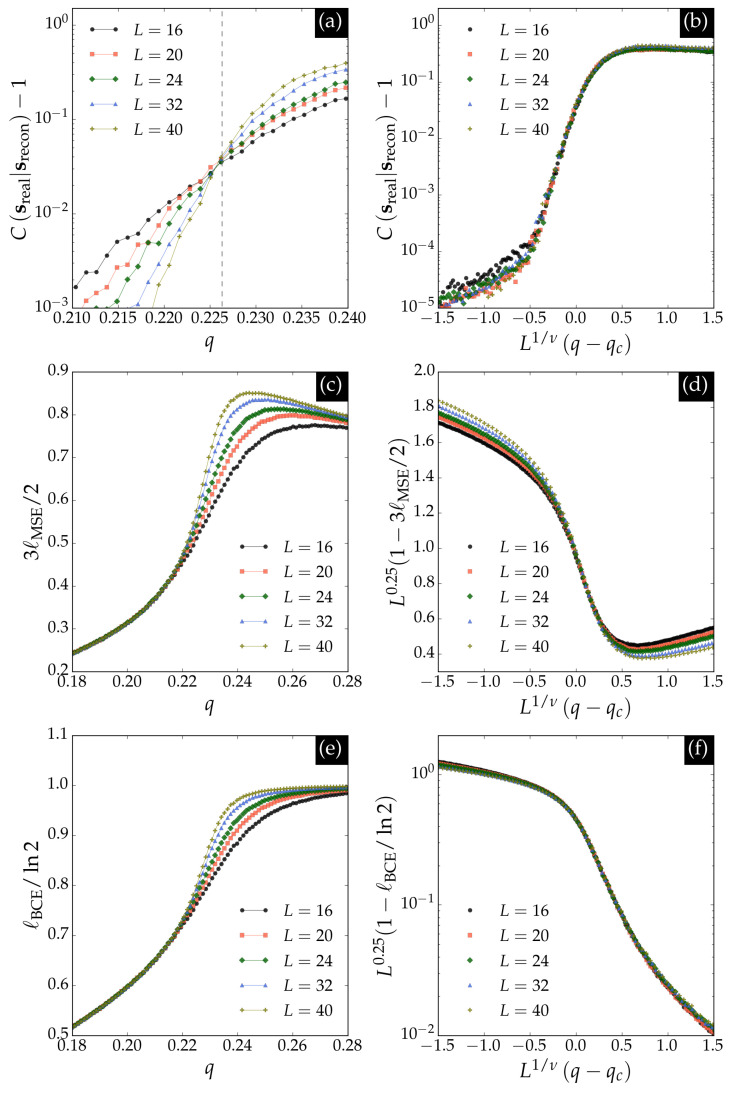
Observables for the reconstructed data by the *VAE*. (**a**) Correlation function from Equation (Equation 12), which is universal at the critical noise $q_{c}^{s}$ (vertical dashed line). (**b**) Scaling collapse of the correlation function according to Equation (Equation 13) with exponent ν=1. Panels (**c**) and (**e**) show $3l_{\mathrm{MSE}}/2$ and $l_{\mathrm{BCE}}/\ln2$, respectively. Panels (**d**) and (**f**) show scaling collapses of $1-3l_{\mathrm{MSE}}/2$ and $1-l_{\mathrm{BCE}}/\ln2$ according to Equation (Equation 15) with exponent 2β/ν=1/4, respectively.

## Data Availability

The data generated and presented in this study are available on reasonable request from the corresponding author.

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
