# Peer review of "Deep Learning of the Biswas–Chatterjee–Sen Model"

_entropy, 2025, doi:10.3390/e27111173_

Round 1
Reviewer 1 Report
Comments and Suggestions for Authors
The authors apply deep learning techniques to the non-equilibrium BCS model. The results confirm, as expected, that these techniques can be used successfully for this type of systems. The paper is well written. I can recommend the publication of this manuscript.
Methods
The authors use deep learning techniques that have been shown in a variety of cases to yield correct results for equilibrium critical systems. They apply these techniques to a non-equilibrium voting model. There is no innovation in techniques, but to my knowledge the current system has not been studied in this way previously.
Results
The aim of this work is to show that the deep learning techniques can be applied to a non-equilibrium system. They study well-known critical quantities and show that hey recover results obtained earlier with traditional numerical simulation methods. The purpose of the paper is not to provide new results, but to show that a rather new numerical technique can be used for this type of systems.
Conclusion
The conclusion of the paper is that deep learning techniques can be used successfully for the current system. This was in fact expected, and any other outcome would have been a surprise.
Author Response
The authors apply deep learning techniques to the non-equilibrium BCS model. The results confirm, as expected, that these techniques can be used successfully for this type of systems. The paper is well written. I can recommend the publication of this manuscript.
We would like to thank the referee for the positive comments and for recommending the publication of our manuscript.
1.1 Methods
The authors use deep learning techniques that have been shown in a variety of cases to yield correct results for equilibrium critical systems. They apply these techniques to a non-equilibrium voting model. There is no innovation in techniques, but to my knowledge the current system has not been studied in this way previously.
The main goal of our work is to develop a metodology based on deep learning techniques to study nonequilibrium systems. As the referee pointed out, the techniques used in our work have been applied to equilibrium systems before, but not to non-equilibrium systems. Thus, our work is innovative in the sense that it extends the application of deep learning techniques to a new class of systems.
1.2 Results
The aim of this work is to show that the deep learning techniques can be applied to a non-equilibrium system. They study well-known critical quantities and show hey recover results obtained earlier with traditional numerical simulation methods. The purpose of the paper is not to provide new results, but to show that a rather new numerical technique can be used for this type of systems.
We agree with the referee’s assessment. The main purpose of our work is to demonstrate that deep learning techniques can be effectively applied to non-equilibrium systems, and we have shown that our results are consistent with those obtained through traditional simulation methods.
1.3 Conclusion
The conclusion of the paper is that deep learning techniques can be used successfully for the current system. This was in fact expected, and any other outcome would have been a surprise.
We appreciate the referee’s recognition of our findings. Our results confirm the effectiveness of deep learning techniques in studying non-equilibrium systems.

Reviewer 2 Report
Comments and Suggestions for Authors
The present submission reports results of the application of certain machine learning techniques to a well-know continuous model of opinion formation on networks. The goal is to identify the critical value of the noise parameter that in some cases is known from analytical approaches, in others can only be estimated from Monte Carlo simulations. Overall, it is shown that both supervised and unsupervised learning techniques perfrom well on this task.
Overall, this is of some interest. The paper is also well-written and succinct so that it appears almost ripe for publication.
I only have tow minor suggestions for clarifications:
- explain wheter updating of opinios happens in diescrete or continuous time
- explain better what format f the data have been used as input in the PCA analysis
Author Response
The present submission reports results of the application of certain machine learning techniques to a well-know continuous model of opinion formation on networks. The goal is to identify the critical value of the noise parameter that in some cases is known from analytical approaches, in others can only be estimated from Monte Carlo simulations. Overall, it is shown that both supervised and unsupervised learning techniques perform well on this task.
We would like to thank the referee for the positive comments and for recognizing the effectiveness of our machine learning techniques in identifying the critical noise parameter in the opinion formation model.
Overall, this is of some interest. The paper is also well-written and succinct so that it appears almost ripe for publication.
We appreciate the referee’s positive view of our manuscript.
I only have two minor suggestions for clarifications:
Explain whether updating of opinions happens in discrete or continuous time.
In the studied dynamics, we updated the agent opinion in discrete time steps. At each time step, one agent updates its opinions based on the current state of the neighbors according Eq. 2 of the manuscript. We store the snapshots of the lattice with the opinions of each agent after a certain number of Monte Carlo steps to ensure that the system has reached a steady state before performing the analysis. We have added this clarification in the Methods section of the manuscript.
Explain better what format for the data have been used as input in the PCA analysis.
In the PCA analysis, we used the snapshots of the lattice containing the opinions of each agent as input data. Each snapshot is represented as a vector, where each element corresponds to the opinion of an agent in the lattice. We separate the stored configurations as functions of the noise parameter.
We also added the reciprocal configurations, i.e., for each configuration at a given noise parameter, we also consider its negative counterpart. This makes the dataset automatically centered around zero.
Finally, we reshaped the data for each noise parameter into an array with dimension (2ND, L2) where ND is the number of configurations. Finally, we passed the resulting array as an input to the pca.fit transform method of Scikit-Learn which implements PCA.
